# Comparison of Motor Evoked Potentials Neuromonitoring Following Pre- and Postoperative Transcranial Magnetic Stimulation and Intraoperative Electrical Stimulation in Patients Undergoing Surgical Correction of Idiopathic Scoliosis

**DOI:** 10.3390/jcm12196312

**Published:** 2023-09-30

**Authors:** Przemysław Daroszewski, Juliusz Huber, Katarzyna Kaczmarek, Piotr Janusz, Paweł Główka, Marek Tomaszewski, Małgorzata Domagalska, Tomasz Kotwicki

**Affiliations:** 1Department of Organization and Management in Health Care, Poznań University of Medical Sciences, 28 Czerwca 1956 r. Str., no. 135/147, 61-545 Poznań, Poland; dyrektor@orsk.pl; 2Department Pathophysiology of Locomotor Organs, Poznań University of Medical Sciences, 28 Czerwca 1956 r. Str., no. 135/147, 61-545 Poznań, Poland; kasia@przystan.pl; 3Department of Spine Disorders and Pediatric Orthopaedics, Poznań University of Medical Sciences, 28 Czerwca 1956 r. Str., no. 135/147, 61-545 Poznań, Poland; pjanusz@ump.edu.pl (P.J.); pawel.glowka@ump.edu.pl (P.G.); mtooma@wp.pl (M.T.); kotwicki@ump.edu.pl (T.K.); 4Department of Palliative Medicine, Poznań University of Medical Sciences, Rusa 55, 61-245 Poznań, Poland; m.domagalska@icloud.com

**Keywords:** scoliosis surgery, pre- and postoperative neurophysiological recordings, intraoperative neuromonitoring, electromyography, electroneurography, motor evoked potentials

## Abstract

The relationships between the results of pre- and intraoperative motor evoked potential recordings during neuromonitoring and whether idiopathic scoliosis (IS) surgical correction improves the spinal efferent transmission have not been specified in detail. This study aims to compare the results of surface-recorded electromyography (EMG), electroneurography (ENG, M, and F-waves), and especially motor evoked potential (MEP) recordings from tibialis anterior muscle (TA) bilaterally in 353 girls with right idiopathic scoliosis (types 1–3 according to Lenke classification). It has not yet been documented whether the results of MEP recordings induced by transcranial single magnetic stimulus (TMS, pre- and postoperatively) and trains of electrical stimuli (TES; intraoperatively in T0—before surgery, T1—after pedicle screws implantation, and T2—after scoliosis curvature distraction and derotation following two-rod implantation) can be compared for diagnostic verification of the improvement of spinal cord neural transmission. We attempted to determine whether the constant level of optimal anesthesia during certain surgical steps of scoliosis treatment affects the parameters of MEPs recorded during neuromonitoring procedures. No neurological deficits have been observed postoperatively. The values of amplitudes but not latencies in MEP recordings evoked with TMS in IS patients compared before and after surgery indicated a slight improvement in efferent neural transmission. The results of all neurophysiological studies in IS patients were significantly asymmetrical and recorded worse on the concave side, suggesting greater neurological motor deficits at *p* = 0.04. The surgeries brought significant improvement (*p* = 0.04) in the parameters of amplitudes of sEMG recordings; however, the consequences of abnormalities in the activity of TA motor units were still reflected. ENG study results showed the symptoms of the axonal-type injury in peroneal motor fibers improving only on the concave side at *p* = 0.04, in parallel with F-wave parameters, which suggests that derotation and distraction might result in restoring the proper relations of the lumbar ventral roots in the spinal central canal, resembling their decompression. There were no significant differences detected in the amplitudes or latencies of MEPs induced with TMS or TES when comparing the parameters recorded preoperatively and intraoperatively in T0. The amplitudes of TES-evoked MEPs increased gradually at *p* = 0.04 in the subsequent periods (T1 and T2) of observation. A reduction in MEP latency at *p* = 0.05 was observed only at the end of the IS surgery. Studies on the possible connections between the level of anesthesia fluctuations and the required TMS stimulus strength, as well as the MEP amplitude changes measured in T0–T2, revealed a lack of relationships. These might not be the factors influencing the efferent transmission in spinal pathways beside the surgical procedures. Pre- (TMS-evoked) and intraoperative (TES-evoked) recordings are reliable for evaluating the patient’s neurological status before and during surgical scoliosis correction procedures. An increase in MEP amplitude parameters recorded on both sides after scoliosis surgery proves the immediate improvement of the total efferent spinal cord transmission. Considering comparative pre- and postoperative sEMG and ENG recordings, it can be concluded that surgeries might directly result in additional lumbar ventral root decompression. We can conclude that MEP parameter changes are determined by the surgery procedures during neuromonitoring, not the anesthesia conditions if they are kept stable, which influences a decrease in the number of false-positive neuromonitoring warnings.

## 1. Introduction

Idiopathic scoliosis (IS) is a developmental deformity of the spine and the trunk in three planes; the most expressed is lateral spine curvature in thoracolumbar vertebrae and rotation along the axis. The results of epidemiological studies show significant incident rate discrepancies from 0.93% to 12% in the world population [1,2]. Untreated scoliosis can lead to significant trunk deformities, changes in the biomechanics of the chest, and the development of internal organ pathologies. Abnormal curvature of the spine, most often quickly developing from the age of four years, affects the anatomical relationships of the spinal cord in the spinal central canal and leads to changes in the activity of the grey matter nerve centers, conduction of nerve impulses in the axons of the lateral funiculi in the white matter, changes in the conduction of spinal roots, development of neuropathy in the peripheral nervous system, and neurogenic changes in the muscular system [3]. In addition to the pathologies mentioned above, the aesthetic factor of a deformed body figure is one of the main reasons for patients and their families to seek the most effective ways of treating IS, which they expect to receive from a spinal surgeon [4].

The conservative treatment by means of physiotherapy [5] and Cheneau-brace [6,7,8] application can be useful for the prevention of scoliotic curve progression and sometimes slows down or limits progression in patients with IS. However, many factors influence the effect of such a therapy, and the surgical implantation of deformity corrective instrumentation is necessary in the majority of progressive IS cases [9], especially when its lateral main angle exceeds 40 degrees [10].

Spinal surgery involves a wide spectrum of procedures, during which the spinal cord, the nerve roots, and the key blood vessels are frequently at risk of injury. Neurologic complications may occur in 6.3% of patients through various mechanisms, including direct trauma to the spinal cord, ischemia, and stretch during IS deformity correction [11]. Intraoperative neuromonitoring provides a safe and useful warning mechanism to minimize spinal cord injury that may arise during scoliosis correction surgery in pediatric patients [12]. This procedure utilizes methods of clinical neurophysiology to assess the afferent and efferent neural impulse transmission in the spinal cord tracts based on the electrical or magnetic stimulation of the sensory and motor pathways [13]. Combined somatosensory-evoked (SEP) and neurogenic motor-evoked (MEP) potential monitoring during IS surgery represents a contemporary standard of care [14] that enables the abandonment of the need for an intraoperative Vauzelle and Stagnar “wake-up” test, which was popular until the end of the 1980s [15]. During critical intraoperative procedures, which may be iatrogenic for the spinal cord structures or its vascularity, the reliable data obtained from neurophysiological recordings are immediately reported to a surgeon, who can then change, repeat, or abandon the last performed procedure [16,17].

The value of motor evoked potential (MEP) recordings in evaluating efferent trans-mission within spinal cord tracts during neuromonitoring associated with spine surgeries is undeniable. However, the vast majority of studies devoted to the surgical correction of idiopathic scoliosis when neuromonitoring procedures have been used describe ambiguously or with little detail MEP parameters that generally should prove the absence of side effects caused by either implant positioning or corrective maneuvers like distraction and derotation. Usually, researchers provided data on the percentages of changes that should be considered critical in intraoperative MEP recordings [18] or focused on selecting the most dangerous elements of the surgical procedure that may affect the occurrence of iatrogenic side effects [19,20]. The morphology and parameters of MEPs recorded intraoperatively either have not been presented, compared, or discussed in detail in the literature, or the relevant studies were performed on a small population of IS patients [21]. In an extensive review of this issue, Chang et al. [22] did not show details of parameter variability but found that during a spinal deformity surgery, combined MEP and SEP monitoring showed high sensitivity and specificity for detecting neural transmission deficits. Most papers are concentrated on the variability of MEPs depending on the number of applied pedicle screws for mounting the corrective spinal instrumentation, manoeuvres, and the type of instrumentation used in IS surgery [20]. Waveform MEP deterioration has been shown to occur commonly during rotation manoeuvres, and more frequently in patients with a larger preoperative lateral spinal curvature. A significant relationship was presented between the number of spinal levels fused and the MEPs’ waveform deterioration [23].

Another problem constitutes the evaluation of asymmetry in the spinal transmission of neural impulses in patients with IS, which seems to be an essential neurological indicator for a surgeon deciding to introduce the treatment at a theatre. Clinical studies usually do not present such a symptom clearly, while functional evaluation with neurophysiological methods reveals subtle but sometimes controversial results. A trend towards increased asymmetries in side-to-side differences in the spinal efferent transmission and cortical latencies was detected, probably representing the subclinical involvement of the corticospinal tracts secondary to mechanical compression, according to the conclusion of Kimiskidis et al. [24]. Luc et al. [25] claim that there is no difference in latencies in MEP examinations of patients with scoliosis on the right and left sides when recorded from the tibialis anterior muscle, which is most often considered the key muscle for neuromonitoring, and is also considered as such in the undertaken work. It seems that the answer to this question may be provided through a comparison of the results of clinical neurophysiology studies in patients with IS, verifying the bilateral efferent transmission of the neural impulses from the upper motor neuron level to the effector (MEP), the conduction of motor impulses in the peripheral nervous system (electroneurography, ENG) and assessing the contractile properties of the muscles themselves (electromyography, sEMG). The paravertebral muscles in patients with scoliosis have been the subject of most electromyographical studies in IS patients [26,27]. At the same time, the effects of disease progression and its surgical and conservative treatment are described in preliminary clinical neurophysiology observations following examining the proximal and distal muscles in the lower extremities [28,29]. In this paper, we describe the results of the studies with the methodology of the MEP recordings using the surface electrodes from the tibialis anterior muscle bilaterally, which is more and more widely used not only in pre- and postoperative diagnostic purposes but also has been proven to be precise enough for intraoperative monitoring in comparison to the standard needle electrodes [30,31]. Our previous pilot results on improving the neuromonitoring methodology [32] are fully compatible with such observations.

It has not yet been documented whether the results of MEP recordings induced by the transcranial single magnetic stimulus can be compared with MEPs induced with the trains of electrical stimuli applied intraoperatively for diagnostic evaluation of the spinal neural transmission. The results provided by Glasby et al. [33] suggest that these measurements may be used comparatively and semi-quantitatively to compare pre-, intra-, and postoperative functional integrity of the spinal cord structures in cases of deformities surgery. It should be, however, remembered that trains of stimuli applied transcranially during neuromonitoring may cause the temporal and perhaps the spatial summation of the efferent impulses to the spinal motoneurones, which are mediated polysynaptically, and therefore, the latency and amplitude parameters of MEPs may show the variability [34]. This study also attempts to determine whether the constant level of optimal anaesthesia during surgical scoliosis treatment affects the parameters of MEPs recorded during neuromonitoring procedures. The results of a study by Lo et al. [35] confirm that MEPs may, rarely, occur unpredictably in susceptible individuals, independent of surgical or anaesthetic intervention. However, they did not provide specific results for the recorded parameters of MEPs.

Is there any relationship between pre- and intraoperative motor evoked potentials’ recordings and does IS surgical correction directly improve spinal efferent transmission? This study aims to compare the results of surface-recorded electromyography (EMG), electroneurography (ENG), and especially motor evoked potentials, not only before and after scoliosis correction but also at three stages of the intraoperative treatment. The review of the literature does not indicate studies on the simultaneous comparison of the MEP results in the same patients treated surgically for idiopathic scoliosis that was recorded pre-, intra-, and postoperatively.

## 2. Materials and Methods

### 2.1. Participants and Study Design

A total of 353 girls with idiopathic scoliosis were included in this retrospective study (Table 1). They were treated surgically at Wiktor Dega Orthopedic and Rehabilitation Hospital in Poznań, Poland.

All the clinical studies before and after treatment (including the analysis of anterior-posterior and lateral X-rays), as well as the surgeries, were performed by the same team of four experienced spine surgeons; neurological status and anaesthesia were evaluated and administered by the same neurologist and anesthesiologist, respectively, for each patient. Two clinical neurophysiologists performed pre- and postoperatively the same set of three diagnostic tests. They comprised (1) bilateral tibialis anterior (TA) muscle electromyography during maximal contraction with surface electrodes (sEMG,) (2) peroneal nerve electro-neurography (ENG) recorded from extensor digitorum brevis (EXT) muscle after electrical stimulation at the ankle, and (3) motor evoked potential (MEP) recordings from tibialis anterior muscles following transcranial magnetic stimulation (TMS). The same neuro-physiological examinations were performed on the group of eighty healthy girls (Table 1) to obtain the reference values for comparison.

Intraoperative neurophysiological monitoring was performed by the same neuro-physiologists, and included recordings of bilateral MEP from muscles of the upper and lower extremities. MEPs were induced following transcranial electrical stimulation (TES). For the purposes of this paper, the results of MEPs from the tibialis anterior muscle are presented, as it is the key muscle most often described in scientific reports for comparison of the results in parameters of neuromonitoring during a scoliosis surgery. Results from the neuromonitoring recordings have been chosen for analysis in T0—the intraoperative observation period before surgery onset, T1— the intraoperative observation period after pedicle screws’ implantation, T2—the intraoperative observation period after corrective rods’ implantation, derotation with the convex rod, apical translation, segmental derotation, dis-traction on the concave side, and compression on the convex side.

Exclusion criteria for TES applied during the neuromonitoring included epilepsy, cortical lesions, convexity skull vault defects, raised intracranial pressure, cardiac disease, proconvulsant medications or anesthetics, intracranial electrodes, vascular clips or shunts, and cardiac pacemakers or other implanted biomedical devices [16].

Ethical considerations were in agreement with the Helsinki Declaration. Approval was received from the Bioethical Committee of the University of Medical Sciences in Poznań, Poland (including studies on healthy people), decisions No. 942/21. Each subject (and her parent/legal guardian) was informed about the aim of the study and gave written consent for examinations and data publication.

### 2.2. Anaesthesia and Spine Surgery

The spine surgeries and recordings of MEPs following trains of the applied transcranial electrical stimulation (TES) were performed under Propofol/Remifentanil anesthesia (induction dose of Remifentanil 0.5 µg/kg and Propofol 2 mg/kg, and later Remifentanil 0.5–2.0 µg/kg/h and Propofol 2–4 mg/kg/h in continuous infusion) with one-time dose of neuromuscular blockade (0.5 mg/kg of rocuronium bromide) at the beginning of the procedure. The level of anesthesia was continuously monitored and ascertained in Bispectral Index Monitor (BIS, GE, Heathcare, Helsinki, Finland); it was kept constant from 40 to 65 during all applied surgery procedures and neuromonitoring MEPs recordings [38]. The arterial blood pressure (kept between 80 and 100), the temperature and %SpO_2_, and CO_2_ partial pressures were continuously monitored and kept within the physiological limits during surgery. Inhalational anesthetics were not routinely applied [39].

In more than half of the patients considered in this paper, the Cheneau-brace or the applied physiotherapy exercises did not bring a significant slowing down of scoliosis progression. At the beginning of the scoliotic spine surgery, the patient was positioned prone on the operation table (Figure 1(Db)). The whole spine was prepped and draped. A posterior midline skin incision was performed. The paraspinal muscles were dissected subperiosteally. The spine was exposed bilaterally from the midline along the spinous processes, laminas to the tip of transverse processes (Figure 1(Cb)). The surgeon cauterized the paravertebral muscles, as the spine was exposed to control bleeding. The spinous processes with supraspinous ligament were preserved for further anatomical wound closure. Removed pieces of the bones from processes and released spine joints were collected and then used as autografts for fine fusion. Pedicle screws were placed bilaterally with free-hand technique (from 8 to 16, 12 on average) (Figure 1(Dd)). All patients had an implanted corrective instrumentation system (Nova Spine, Amiens, France). Polyaxial and monoaxial screws with 5.5 mm rods made of titanium alloy were used (Figure 1(Cc,De)). The deformity was corrected by combining the following manoeuvers: convex rod rotation, apical translation, segmental derotation, distraction on the concave side and compression on the convex side (Figure 1(Df)). To obtain spine fusion, decortication was performed, and locally harvested bone grafts were used. The wound was closed in layers over a subfascial drain. The location, angle, and depth of the pedicle screw implantation were controlled with the X-ray C-arm for intraoperative imaging (Figure 1(Da)).

### 2.3. Neurophysiological Recordings

Figure 1 presents the methodological principles of the neurophysiological studies. The examinations were performed in an air-conditioned room with a controlled temperature of 22 °C. Surface electromyography (sEMG) recordings were performed bilaterally from the tibialis anterior muscle before and after the surgery to evaluate the motor unit recruitment during the attempt of a 5 s maximal contraction (Figure 1(Bc)). The sEMG recordings were performed using the KeyPoint Diagnostic System (Medtronic A/S, Skøvlunde, Denmark) with patients in a supine position during the examination. For measurements, we applied standard, disposable Ag/AgCl surface recording electrodes (5 mm^2^ of an active surface) with an active electrode placed on the muscle belly, a reference electrode placed on the distal tendon of the same muscle, and a ground electrode placed on the distal part of the examined muscle, according to the Guidelines of the International Federation of Clinical Neurophysiology—European Chapter [40,41,42,43,44]. Patients were instructed to contract the muscles under examination and make the strongest possible contraction of the muscles for 5 s. Three attempts were performed each time. The neurophysiologists selected the best attempt for analysis independently—the one with the highest mean amplitude measured peak-to-peak with reference to the isoelectric line. The output measures were the amplitude measured in μV and the frequency of muscle motor unit action potential recruitment measured in Hz. A frequency index (FI, 3–0) was scored based on the calculations of motor unit action potential recruitment during maximal contraction in sEMG recording: 3 = 95–70 Hz—normal; 2 = 65–40 Hz—moderate abnormality; 1 = 35–10 Hz—severe abnormality; 0 = no contraction. sEMG recordings in both controls and patients were performed at a base time of 80 ms/D and an amplification of 20–1000 μV/D. We set the upper 10 kHz and the lower 20 Hz filters in the recorder.

Bilateral electroneurography (ENG) was performed to assess the transmission of neural impulses in the motor peripheral fibres of the peroneal nerves. The aim was to assess whether there are significant differences in nerve conduction that can negatively affect the evaluation of the muscle function or the efferent transmission measurements. The procedure involved delivering rectangular pulses of 0.2 ms duration at a frequency of 1 Hz and an intensity ranging from 0 to 80 mA using bipolar stimulating electrodes placed over the skin along the anatomical passages of the nerves at the ankle. The compound muscle action potentials M-waves (CMAP) and F-waves were recorded from the extensor digitorum brevis muscles (EXT). Recordings of these potentials verified transmission of neuronal impulses in the motor fibres peripherally and within L5 ventral spinal roots, respectively. The recordings were performed at the amplification of 500–5000 µV/D and a time base of 5–10 ms/D, and compared to normative values recorded in the healthy volunteers with the patients. The outcome measures were the parameters of amplitudes (in µV) and latencies (in ms) for M–waves, interlatencies of recorded M-F waves (in ms), and frequencies for F-waves (usually not less than 14 during evoking 20 positive, successive recordings of M—waves). The measurements were performed at an amplification of 5–5000 µV and a time base of 2–10 ms. The normative values recorded in healthy volunteer subjects were then compared with the test results of the patients. More details on the methodology of acquisition and interpretation of ENG studies are described in other papers published by our team members [41,42].

Motor evoked potentials (MEPs) were elicited using transcranial magnetic single stimulus (TMS, biphasic, 5 ms lasting) using a magnetic circular coil (C-100, 12 cm in diameter) placed over the scalp in the area of the M1 motor cortex targeted with an angle for the corona radiate excitation, where the fibres of the corticospinal tract for upper and lower extremities originate (Figure 1(Bb)), and recorded with surface electrodes from TA muscles bilaterally (Figure 1(Bc)). The MagPro X100 magnetic stimulator (Medtronic A/S, Skovlunde, Denmark) was used for the MEP testing. The magnetic field stream delivered from the coil at the strength 70–80% of the resting motor threshold (RMT; 0.84–0.96 T) excited all neural structures up to 3–5 cm deep. We can suppose that with such a condition, the cells of origin of the rubrospinal tract in the midbrain are also excited. The latency and amplitude parameters were analyzed as the primary outcome measure to assess the primary motor cortex output and evaluate the efferent transmission of neural impulses to effectors via spinal cord descending tracts (Figure 1A). Attempts of consecutive trackings searched the optimal stimulation location (a hot spot in the area where TMS elicited the largest MEP amplitude, Figure 1(Ba)) distanced 5 mm from each other. The amplitude was measured from peak to peak of the signal, the latency from the stimulus application marked by the artefact in the recording to the onset of the positive inflexion of potential. The patients and healthy volunteers did not report the stimulation as painful, but they felt the little current spread to the lower extremities. They were always awake and cooperating. MEPs were recorded using the 8-channel KeyPoint Diagnostic System (Medtronic A/S, Skøvlunde, Denmark). Standard disposable Ag/AgCl surface electrodes with an active surface of 5 mm^2^ were used. The ground electrode was located on the leg, near the knee. The recorder’s low-pass filter was set to 20 Hz, high-pass filter to 10 kHz, and the time base at 10 ms/D, the amplification of signals was set between 200 and 5000 µV. A bandwidth of 10 Hz to 1000 Hz and digitalization at 2000 samples per second and channel were used during recordings. The resistance between the electrode surface and the skin was decreased with electroconductive gel. The methodology of MEP recordings has been described in detail elsewhere [40,41,42,43,44].

Neuromonitoring sessions were performed in the theatre at the same temperature of 22 °C using the ISIS system (Inomed Medizintechnik, Emmendinger, Germany) (Figure 1(Dc)). Motor evoked potentials were induced as a result of transcranial electrical stimulation (Figure 1(Ab)) in areas of the cortical motor fields for innervation of the thumb and selected muscles of the lower extremities (Figure 1(Ca)) through a sequence of four stimuli (duration of a single pulse 500 µs) with an intensity of 40–170 mA via bipolar subcutaneous electrodes. Stimulating electrodes were positioned over the skull according to the 10-system: Cz–C3 3–6 cm to the left, and Cz to C4 3–6 cm to the right following the compilation of description by Deletis [13] and Legatt et al. [45]. The impendence of scalp electrodes was about 0.8 kΩ. Particular attention was paid to ensure that the level of anaesthesia (indications of BIS) and the strength of electrical stimuli (in mA) adjusted at the beginning of the surgery did not significantly change and were maintained at the appropriate level throughout the applied corrective procedures. The needle ground electrode was applied in the area of the iliac crest. We used our experience in the utilization of the surface electrodes for MEP recording from TA muscles according to the previous descriptions [32]. Their impedance measured at the beginning of the neuromonitoring sessions was 10–20 kΩ. The recorded potentials were characterized by a variable amplitude from 100 to 2000 µV and latencies in the range of 27 to 40 ms, depending on the conduction distance. The potentials did not require averaging. The following standard settings of measurements were applied: filters hardware high-pass [Hz] 30; software high-pass [Hz] 0.5; software low-pass [Hz] 2000; stimulation frequency [Hz] at 0.5–2.4 ms intervals. Before starting the surgery, after implanting the stimulating (Figure 1(Ca)) and recording (Figure 1(Cd)) electrodes in the supine position of the patient (Figure 1(Db)), the electrodes’ impedances were checked; the correct values for needle electrodes (Figure 1(Ce)) were in the range of 0.1 to 5.0 kΩ, indicating proper connections with the recorder’s amplifier.

After the patient was transferred to the operating table in the prone position, the MEPs with reference amplitude and latency values were recorded (reference values, T0) for comparison with those that were recorded in the subsequent stages of the surgery (T1 and T2). Amplitudes (in µV) and latencies (in ms) of MEPs were the outcome measurements. All results of MEPs obtained in patients intraoperatively were also compared to the preoperatively recorded results following the magnetic stimulation, aiming to verify the compatibility of the patient’s neurophysiological status regarding the neural efferent impulses transmission. Neuromonitoring was carried out at every stage of surgical correction of scoliosis, and each change in the amplitude or latency parameter of the recorded MEP induced by TES and recorded bilaterally from the muscles of the upper (abductor pollicis brevis) and lower (rectus femoris, tibialis anterior, and abductor hallucis) extremities was reported to the surgical team. For this paper, the results of MEP parameters recorded from the tibialis anterior muscle are presented. A list of the most common reasons for such fluctuations was selected, and their frequencies were calculated. For example, pilot observations indicated that overheating of the tissues accompanying the cauterization before T1 could affect the conduction of nerve impulses in the spinal cord pathways within the white matter funiculi. The surgeon was warned in these cases, and the surgical area was rinsed with the 0.9% NaCl solution at 36.6 °C. Surprisingly, this symptom retreated after the suction of the fluid was applied.

Calculations were made on how often such activity caused the latency parameter fluctuation in MEPs recorded from the anterior tibial muscles.

### 2.4. Statistical Analysis

Data were analyzed with Statistica, version 13.1 (StatSoft, Kraków, Poland). Descriptive statistics were reported as minimal and maximal values (range), with mean and standard deviation (SD). The normality distribution and homogeneity of variances were studied with the Shapiro–Wilk test, and the homogeneity of variances were studied with Levene’s test. The frequency sEMG index, recorded F-wave frequencies, and BIS data were of the ordinal scale type, while amplitudes and latencies were of the interval scale type. However, they did not represent a normal distribution; therefore, the non-parametric tests had to be used. None of the collected data represented a normal distribution or was of the ordinal scale type; therefore, the Wilcoxon’s signed-rank test was used to compare the differences between results obtained before and after surgeries, as well as to compare results at the beginning (T0), during (T1) and in the end (T2) of the surgical procedures. In the cases of independent variables, the non-parametric Mann–Whitney test was used. Any *p*-values of ≤0.05 were considered statistically significant. The cumulative data from parameters of MEP recordings performed on both sides were used for comparison of the relationships between BIS read-outs in T0, T1, and T2. The results from all neurophysiological tests performed on patients were also calculated from the group of healthy subjects (control group) to achieve the normative parameters used to compare the health status between the patients and the controls. Results did not reveal any significant difference in values of parameters recorded in neurophysiological tests on the left and right sides in controls. Attention was paid to matching patients and healthy controls’ demographic and anthropometric properties, including gender, age, height, weight, and BMI. Statistical software (StatSoft, Kraków, Poland) was used to determine the required sample size using the primary outcome variable of sEMG and MEPs amplitudes recorded from TA muscles before and after treatment with a power of 80% and a significance level of 0.05 (two-tailed). The mean and standard deviation (SD) were calculated using the data from the first hundred patients, and the sample size software estimated that more than two hundred patients were needed for this study.

## 3. Results

During neuromonitoring in T0, the impedance of the stimulating electrodes distributed with the 10–20 systems inserted under the skin over the skull was 0.8 ± 0.2 kΩ. The impedance of the surface disposable bipolar recording electrodes from muscle groups was in the range of 10 to 20 kΩ (mean of 13.2 ± 1.3 kΩ).

The coincidence of the proper positioning of the electrodes stimulating the transcranial motor centres for the innervation of more lower than the upper muscles using measurements of the 10–20 system with the method of determining the “hot spots” during the recording of the largest amplitude preoperative MEP was calculated at 86%.

Data in Table 2 indicate that during surgeries, the events evoking the fluctuation of intraoperatively recorded MEPs parameters (more amplitudes than latencies) and reported to the surgeons were associated the most frequently with pedicle screw implantation, corrective rod implantation, derotation with a convex rod, and distraction on the concave side. Heating of the spine related to cauterization was the most frequent reason for latency fluctuation in the MEP evoked TES recordings. Among 353 neuromonitoring cases described in this paper, none of the listed incidents reported to surgeons with immediate reactions resulted in a significant postoperative neurological or motor function deficit.

Data on parameters of sEMG and ENG recordings indicate (Table 3) that muscle motor units’ activity and conduction of the motor impulses in peroneal nerve fibers peripherally in IS patients were significantly different from the healthy controls before and after the surgery. The difference in MEPs’ amplitudes before surgery (Figure 2E) was at *p* = 0.009 bilaterally. After the treatment (Figure 2G) it was at *p* = 0.02 − 0.01, indicating a slight improvement in the efferent transmission of neural impulses with the fibres of the spinal tracts postoperatively.

Preoperatively, the results of all neurophysiological studies in IS patients (Figure 2E,(Fa),H) were significantly asymmetrical and recorded worse on the concave side, suggesting more significant neurological motor deficits at *p* = 0.04. One week postoperatively, this asymmetry was recorded as significantly reduced (Figure 2(Fb),I).

The surgeries in IS patients brought a significant increase in amplitudes at *p* = 0.04 but not FI in sEMG recordings, bilaterally (Figure 2H,I; upper traces), which points to the improvement in the activity of muscle motor units still with the signs of the neurogenic type of abnormality. The decreased values of M-waves amplitudes and latencies recorded in ENG examinations (Figure 2H,I; bottom traces) indicated the symptoms of peroneal motor fibre injury of the axonal type, and improved only on the concave side at about *p* = 0.04. They were in parallel with the significant increase in the values of F-waves parameters (*p* = 0.04), which suggests that surgeries might improve the lumbar ventral roots’ neural transmission. During ENG stimulation studies, the strength of the current to evoke the maximal M-wave in healthy volunteers ranged from 18 to 40 mA with a mean of 27.7 ± 2.4 mA, while in patients it ranged from 38 to 65 mA (mean of 43.7 ± 2.6 mA) preoperatively and from 32 to 63 mA (mean 42.9 ± 2.2 mA) postoperatively.

No significant differences were detected in the amplitudes or latencies of MEP induced with TMS or TES compared to the parameters recorded preoperatively (one day before surgery) and intraoperatively in T0. The amplitudes of TES-evoked MEPs increased gradually at *p* = 0.04 in the subsequent periods (T1 and T2) of observation. The significant reduction in MEP latency at *p* = 0.05 was observed only at the end of the IS surgery.

The total time of the surgical procedures, from transferring the patient to the operation table in a prone position to the final suturing of the wound over the surgical field, ranged from 4.5 to 5.5 h (5 h on average). The additional half an hour should be added to consider the total time of the patient’s anaesthesia.

Particular attention was paid to ensuring that the level of anaesthesia (BIS indications) and the strength of electrical stimuli (in mA) did not significantly change and were maintained at the same level throughout the neuromonitoring procedure. Preliminary studies on the possible relationships between the level of anaesthesia fluctuations and required TMS stimulus strength, as well as the MEP parameter changes measured in T0–T2 periods of observations, were performed in 40 patients undergoing scoliosis surgeries (Figure 3).

The value of the electrical stimulus strength for evoking the highest and most stable MEP amplitude parameter was attempted to be kept constant, and its value ranged from 80 to 130 mA (mean of 97.6 ± 12.4 SD) (Figure 3B).

The average value of the BIS parameter measured during about five hours of the surgery was 56.5 ± 4.8 at the beginning of the scoliosis correction procedure (T0), a value which slightly decreased to 55.3 ± 3.7 in the middle of the procedure (T1), and reached 58.1 ± 3.0 after its completion (T2), which may suggest that the changes in the anaesthesia level applied to the patients were only discrete (Figure 3A). These differences were not statistically significant (at *p* = 0.09). It should be remembered that the difference in the range of 10 in BIS measurements is clinically insignificant.

With the same periods of observation, the cumulative mean values of the MEPs amplitude parameter recorded from the anterior tibialis muscles were 409.0 ± 58.5 µV (T0), 406.6 ± 76.5 µV (T1), and 562.5 ± 45.9 µV (T2), respectively. The difference between recordings at T0 and T2 was statistically different at *p* = 0.03.

The cumulative mean values of MEP latencies recorded in T0 were 32.0 ± 2.0 ms, 32.9 ± 2.2 ms in T1, and 32.7 ± 2.1 in T2, and the differences between them were statistically insignificant (at *p* = 0.21 and *p* = 0.35). There were no significant relationships between BIS fluctuations (Figure 3A) and the applied electrical stimulus strengths (Figure 3B) for evoking the maximal MEP amplitude trends at three observation periods.

The above data may suggest an Improvement in the spinal conductivity of neural impulses but a lack of relationship between the fluctuation of the MEP amplitude parameter and the applied level of anaesthesia (Figure 3A) or the constant electrical stimulus strength (Figure 3B) during surgeries of patients with IS under this study. It is not likely that they could be the factors influencing the efferent transmission in the spinal pathways bilaterally recorded in MEP tests besides the surgical procedures.

## 4. Discussion

The results of MEP recordings evoked with TMS in this study indicated a slight improvement in the efferent transmission of neural impulses within the fibres of the spinal tracts in IS patients postoperatively. The results of all neurophysiological studies were significantly asymmetrical and recorded worse on the concave side; this asymmetry was significantly reduced following IS surgery. The surgeries in IS patients significantly improved the parameters of sEMG recordings; however, they still reflect the consequences of the neurogenic abnormal activity of TA muscle motor units. ENG studies results proved the axonal-type injury symptoms in peroneal motor fibres, which postoperatively improved only on the concave side in parallel with the lumbar ventral roots motor conduction. MEP parameters induced with TMS preoperatively and TES at T0 did not differ. The amplitudes of TES-evoked MEPs increased gradually in two periods of intraoperative observation (T1 and T2). Studies on the possible influence between the level of anaesthesia and fluctuations in MEP amplitudes did not reveal a direct relationship.

The compatibility between the positioning of the electrodes stimulating the motor centres transcranially for the innervation of lower rather than upper muscles using the 10–20 system measurements with the method of determining the “hot-spots” during preoperative MEP recordings was calculated at 86%. This variability is partly due to the human individual differences in the distribution of motor centres [45], which was also reported in their pioneering works by Penfield and Jasper [46] as “paradoxical distribution of motor centres”. This suggests that the 10–20 method should be routinely combined and compared preoperatively with MEPs induced with “hot spots” to avoid complications during neuromonitoring in the theatre at T0, which was underlined by Garcia et al. [47]. The same applies to the general idea of preoperative neurophysiological tests performed each time in treated patients with IS, enabling the accurate recognition of changes in efferent neural transmission through MEPs recordings and the functional ability of muscle motor units to a contraction in non-invasive sEMG recordings. They also include the recognition of the degree of asymmetry of the recordings and the level of neuromere in which there are the most significant deficits in the activities of the motor centres [48,49].

Similarly to Gadella et al. [30] and Duffler et al. [31], we observed in T0 twice as many incidents of impedances of surface electrodes than needle electrodes, which did not significantly influence the signal-to-noise ratio parameter, and demonstrated the high utility of both methods in neuromonitoring procedures. Our previous pilot results on improving the neuromonitoring methodology [32] are fully compatible with their observations. Moreover, taking into account the fact that IS surgeries are pediatric and the consequences of neuromonitoring procedures using TES when MEPs are recorded with needle electrodes can be ecchymosis and bruises associated with the stimulation-related muscle movements, local nerve damage or infections in rare cases [50], and, frequently, postoperative skin reddening [51], recording from the muscle’s surface is more beneficial.

According to data from Wang et al. [52], anaesthesia can significantly affect the reliability of TES-evoked MEP monitoring. The results of our preliminary studies on the possible variability of the anaesthesia level on the parameters of intraoperative recorded MEPs on 40 patients show no clear relationship. We can conclude that during our recordings, MEP parameter changes are determined by the surgery procedures during neuromonitoring, not the anaesthesia conditions if they are kept stable, which influences a decrease in false-positive warnings. Our study did not confirm anaesthesia-related warnings as frequent during spinal deformity surgery, contrary to the result reported by Acharya et al. [53], when 50% of the alerts were associated with anaesthetic management.

The contemporary studies of MEP recordings in IS patients assessing the pathologies in the efferent transmission or the effectiveness of treatment provide slightly different values of latencies recorded from lower extremity muscles compared to our results. This may be due to the multiple routes of excitation within anatomical structures in the supraspinal and spinal systems to motoneuronal centres involving di- or trisynaptic pathways, giving a delay difference of 3–4 ms (Figure 1A) or the consequences of summation of the excitatory neural impulses in efferent pathways evoked by the trains of transcranial electrical stimulation. The apparent reason is the difference in the conduction distance influencing the MEPs latency parameter following TES to the recording site in lower extremities muscles, both in the population of IS patients (with different angles of primary or secondary curvatures) and healthy controls of different ages and heights range [54]. However, the results of MEPs parameters recorded preoperatively following TMS in this study are very similar to MEPs induced with TES, which leads to the conclusion that both methods are comparable in the sensitivity and reliability of the assessments. In conclusion, we believe that discrete transient changes in the latency during the whole surgery, and detected especially in T1 as probable side effects and reported to the surgeon, although statistically insignificant, were more related to the heating from the cauterization, or other technical reasons, and not due to a pathophysiological reason, although the exact reason, at present, remains unknown. Our study, similar to the findings of Toki et al. [55], pointed to the lack of statistically significant difference in the MEP latency parameters following the application of single versus trains of transcranially applied impulses.

The comparison of the amplitude parameter of the MEPs recorded in this study from the TA muscle with the reports of other authors is different [22,24]. Lo et al. [56] reported a consistent average latency parameter of about 31–32 ms, but an average amplitude parameter of about 46.5 µV, ten times lower than that presented in the current study, assuming the same type of anesthesia used in patients with IS. On the other hand, Edmonds et al. [21], following nitrous oxide narcotic anaesthesia application during the onset of the surgery of twelve IS patients, reported similar results to those presented in this study, with the mean parameter of amplitude being 490 µV and the latency being 32.0 ms. Suppression of the anaesthesia level diminished more than half of the amplitude parameter in their study.

Analyzing the data listed in Table 2 on sources of evoking the fluctuation of intraoperatively recorded MEPs parameters, their interpretation of the mechanism of action could be proposed. Pedicle screw implantation may cause mechanical spine bending along its axis, causing pressure on the paravertebral vessels or direct pressure on the lateral and ventral spinal cord funiculi structures. Moreover, it can be a source of stretching the bone structure of the vertebral body by the screw or occasionally direct pressing to the nerve structures. Corrective rod implantation can cause the compression of a deformed spinal cord, which has can occur in addition to physiological curvatures during the pathological lateral curvature progression and rotation in the ontogenesis. The correction of spinal deformity, causing a greater frequency of the surgeon warnings by the neurophysiologist during neuromonitoring, should be considered the most dangerous stage of scoliosis surgery, which supports the similar conclusion by Morota et al. [57] and Dormans et al. [58]. Waveform deterioration occurs more frequently during rotation manoeuvres in patients with a larger preoperative lateral scoliosis angle [23].

One of the possible explanations for immediately improving the total efferent transmission revealed in the changes of sEMG, ENG and, MEP parameters following IS spine surgery may be restoring the correct anatomical and functional relationships of the nervous structures in the central canal of the deformed spine. This applies to the axons in the lateral and ventral white matter funiculi and especially to the spinal roots. The ENG study’s results indicated the symptoms of axonal-type injury in peroneal motor fibres, improving only on the concave side at *p* = 0.04 in parallel with the significant improvement of F-waves parameters, which suggests that derotation and distraction might result in restoring the proper relations of the lumbar ventral roots in the spinal central canal, resembling their decompression. Although the surgeries during the scoliosis correction were performed in the lower spine at L1–L3 vertebrae, due to the phenomenon of the pseudoascending of the spinal cord (including the cauda equina), they could also affect those roots at the L5 level that are related to the innervation of the TA muscle.

One of the study’s limitations could be the selection for the final analysis of only the MEP recordings from TA muscle, although data from the other proximal and distal muscles of lower extremities bilaterally were also collected. It is accepted that during the neuromonitoring procedure, the MEP recordings from the rectus femoris, TA, calf group, and abductor hallucis longus muscles provide the highest sensitivity and specificity and best predictive power for postoperative lower extremity weakness [59]. However, the numerical data of other researchers are rarely offered for recordings from these muscles. Therefore, we have chosen TA because the MEPs’ monitoring data are the best accessible for comparison in the literature.

The results presented in this study for the first time provide evidence of the possibility of using pre-, intra-, and postoperative MEP recordings as an effective and accurate tool for predicting and detecting the probable neurological deficits during spine surgery. Moreover, our prospective study seems to fill the gap in validating protocols to manage functional evaluation with neurophysiological methods on specific steps of IS patients’ treatment [60]. In recent years, many spine surgeons have advocated MEP monitoring for all spinal surgery since it better predicts good postoperative motor outcomes than using SEP alone. Moreover, patients with immature neural pathways or preexisting neuromuscular disease may have abnormal baseline SEP recordings, which refers to IS patients [61]. Transcranial electric motor evoked potentials are exquisitely sensitive to altered spinal cord blood flow due to either hypotension or a vascular insult. Moreover, changes in transcranial electric motor evoked potentials are detected earlier than in somatosensory evoked potentials, facilitating a more rapid identification of impending spinal cord injury [62].

Our results confirm Lo et al.’s observation [35] that MEP abnormalities may rarely occur unpredictably, independent of surgical or anaesthetic intervention. Moreover, they also support the necessity of the preoperative MEP recordings presented in this study, because the early recognition of their parameters is essential to prevent false positives during IS spinal surgery. In addition to the analysis of the pre- and postoperative recording results provided in the lower part of Table 2 (as well as in Figure 2E,G), which support that the TMS-induced MEP latencies do not change significantly and only the parameter of bilateral amplitudes increases postoperatively; it can be mentioned that peak latencies were occasionally delayed in postoperative recordings. This might be due to the increase in the MEP duration (dispersion) when recorded in some postoperative patients. Despite careful marking of the MEPs’ onset cursor from the moment of its positive component increase with reference to the isoelectric line, the values of the latency parameter remained generally unchanged during the pre- and postoperative observation periods. However, the morphology (number of components) of the recorded MEPs has changed. The increase in the evoked potential duration is usually interpreted as caused by the aggravation of the pathological factor in the neural conductivity of the nerve or the spinal funicular fibres. In our study, it might be caused by a mechanical factor, evoking a physiological or pathophysiological effect, which may include the impact of corrective instrumentation implantation in the form of two rods made of titanium alloy or the spinal structures distraction effects following the surgical procedures, or a factor that has not yet been revealed. We believe that observations in the more extended postoperative period with MEP control recordings will at least partly clarify whether this is a permanent or transient phenomenon.

The many advances in motor system assessment achieved in the last two decades undoubtedly improved monitoring efficacy without unduly compromising safety. Further studies and experience will likely clarify existing controversies and bring new advances [14]. Motor evoked potentials are the modality of choice for monitoring motor tract function; complete neuromuscular blockade makes their utilization questionable [61]. The future of developing neuromonitoring methods with MEP recordings should consider not only non-invasive methods with surface electrodes, but also studies exploring the approach of nerve- versus muscle-recorded MEP [63]. These are particularly important, considering the “resistance” of nerve-recorded potentials to paralysis applied by anesthesiologists during the intraoperative neuromonitoring of spine surgeries [64].

In terms of basic research, especially an attempt to explain the etiopathogenesis of IS, the results presented in this study show how important the impact of the asymmetry and abnormalities of spinal neural transmission may be in the main curvature progression. It appears to be a pathology secondary to a primary cause located at the supraspinal level [65,66]. The neurological origin of IS development is still only the hypothesis, although the results of this study may shed light on it. However, they do not authorize us to draw broader conclusions. Improving the parameters of MEPs after correction of scoliosis might not only be due to the change in the efferent neural transmission, but many other spinal cord mechanisms could be involved, providing the asymmetry of the tendon reflexes recorded in the paraspinal muscles mainly on the convex side, reflecting the changes in the motoneuronal excitability, possibly involving other neuromeres of the scoliotic spinal cord [67]. The mechanical correction of the scoliosis may change this mechanism.

The clinical significance of the presented study is mainly related to the possibility of the precise assessment of the surgical treatment results using functional tests of clinical neurophysiology and forecasting the need for further surgical treatment associated with the natural progress of patients’ height.

## 5. Conclusions

Considering that MEPs’ amplitude parameter reflects the number of axons excited from the motor cortex and transmitting the efferent impulses via spinal descending tracts in the white matter, pre- (TMS evoked) and intraoperative (TES evoked) recordings are reliable for evaluating the patient’s neurological status before and during surgical scoliosis correction procedures. The results of this study indicate an agreement between preoperative and early intraoperative evaluations with both diagnostic methods. An increase in MEP amplitude parameters recorded on both sides after scoliosis surgery proves the immediate improvement of the total efferent spinal cord transmission. Considering comparative pre- and postoperative sEMG and ENG recordings, it can be concluded that surgeries might directly result in additional lumbar ventral root decompression.

The results of our tests on the possible variability of the anaesthesia level on the parameters of intraoperative recorded MEPs show no clear relationships. We can conclude that the surgery procedures determine MEP parameters’ changes during neuromonitoring, not the anaesthesia conditions if they are kept stable, which influences a decrease in false-positive neuromonitoring warnings.

Further studies on a large population of patients with long-lasting observation postoperatively are required to confirm the presented conclusions on the direct influences of scoliosis surgery on the improvement of motor function in patients with IS. The use of intraoperative neuromonitoring in IS surgery, often complicated due to the possible neurological deficits, not only offers safety for the patient, but also protects the hospital managers from possible consequences due to the patient’s claims.

## Figures and Tables

**Figure 1 jcm-12-06312-f001:**
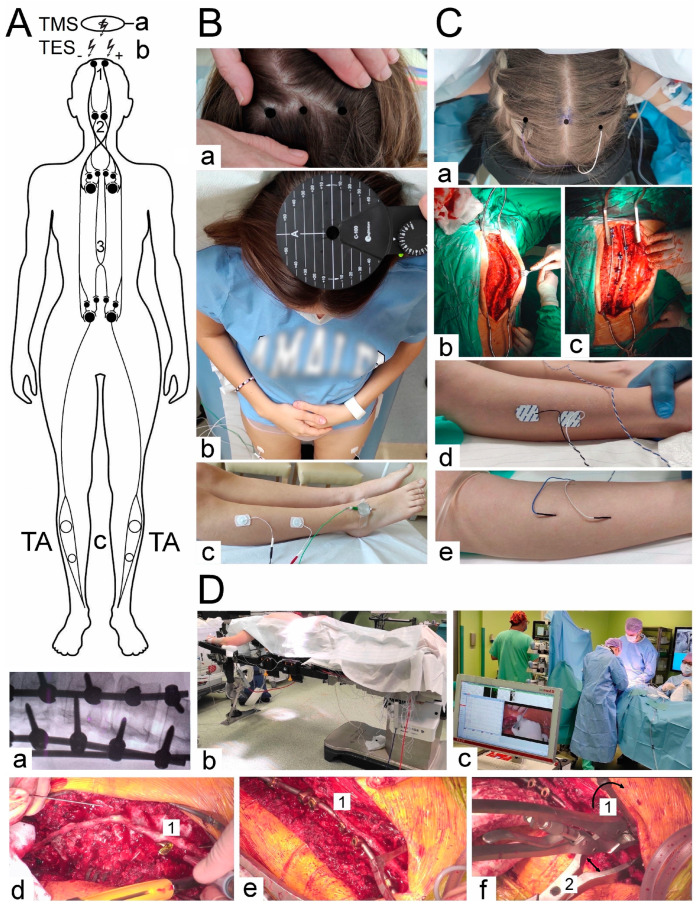
(**A**) A simplified diagram of the anatomical structures transmitting the neural excitation to the motor centres of the spinal cord after TMS (a) or TES (b) of the motor cortex centres. Large black circles denote motoneurones, medium-sized—cells of origin of the descending efferent pathways, small—interneurones. Open white symbols show location of bipolar surface electrodes for MEPs and sEMG recordings from TA muscles bilaterally (c). ⤚—excitatory synapses. 1—corticospinal tract, 2—rubrospinal tract, 3—long descending propriospinal tract. (**B**,**C**) Photographs illustrating methodology of the pre- and postoperative MEPs (**Bb**) and sEMG (**Bc**) recordings with pairs of electrodes placed bilaterally over the surface of TA in healthy volunteers and in patients with scoliosis. “Hot spot” stimulating points were detected and marked preoperatively (**Ba**) following TMS (**Bb**) for TES (**Ca**) purposes performed intraoperatively with needle electrodes and recorded from TA with surface electrodes (**Cd**) or occasionally with needle electrodes (**Ce**). (**Cb**)—a view of the thoracolumbar spine prepared before the scoliosis correction. (**Cc**)—two implanted rods for distraction and derotation procedures of the scoliosis correction. (**D**) (**Da**)—intraoperative X-ray coronal image of the thoracic spine with the implanted screws to the vertebrae and two rods. (**Db**)—a view of the patient in the theatre in the prone position with the prepared back area for the surgical approach. (**Dc**)—a view of the neuromonitoring device in the theatre with the distance from the surgery table. Certain steps of the scoliosis surgery: (**Dd**)—pedicle screw (1) implantation, (**De**)—corrective rod (1) implantation, (**Df**)—correction manoeuvres, derotation (1) and distraction (2) of the spine curvature. Abbreviations: TMS—transcranial magnetic stimulation; TES—transcranial electrical stimulation; TA—tibialis anterior muscle; MEP—motor evoked potential; sEMG—surface electromyography; A-P—anterior-posterior.

**Figure 2 jcm-12-06312-f002:**
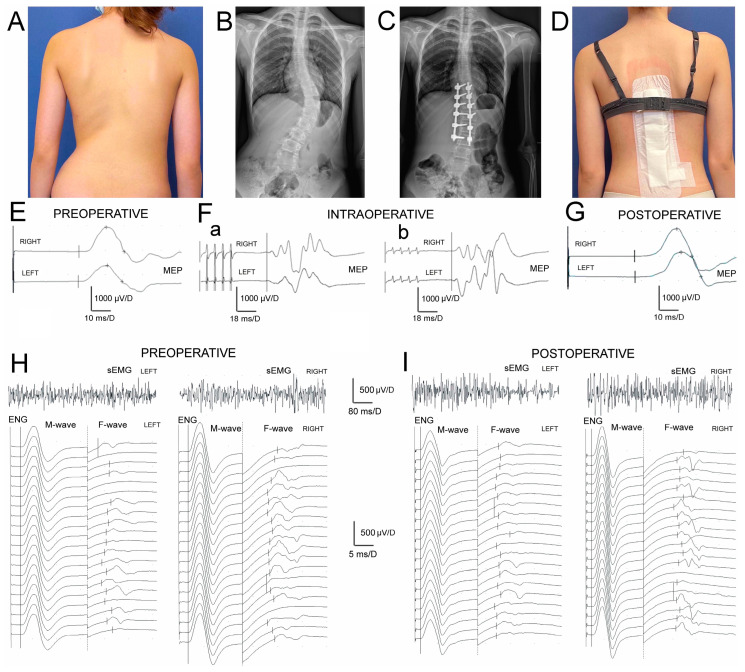
(**A**–**D**) Photographs of patients treated before (**A**,**B**) and after (**C**,**D**) for Lenke type 2 right-sided thoracic and left-sided thoracolumbar scoliosis and the anteroposterior X-rays before and after surgery. Implanted corrective instrumentation is visible on the X-ray in (**C**). Bilateral TMS-induced MEP recordings in the pre- and postoperative evaluation are shown in (**E**) and (**G**), respectively. (**F**,**G**) show TES-induced intraoperative MEP recordings in T0 and T2 follow-up periods, respectively. The stimulation artefact of the TES-induced MEPs (**F**) in the case of this patient whose recordings are presented alters because the stimulation intensity was adjusted from 95 mA (**F**(**a**)) to 85 mA (**F**(**b**)). In (**H**,**I**), bilateral sEMG and ENG recordings are shown in the pre- and postoperative periods for comparison, respectively.

**Figure 3 jcm-12-06312-f003:**
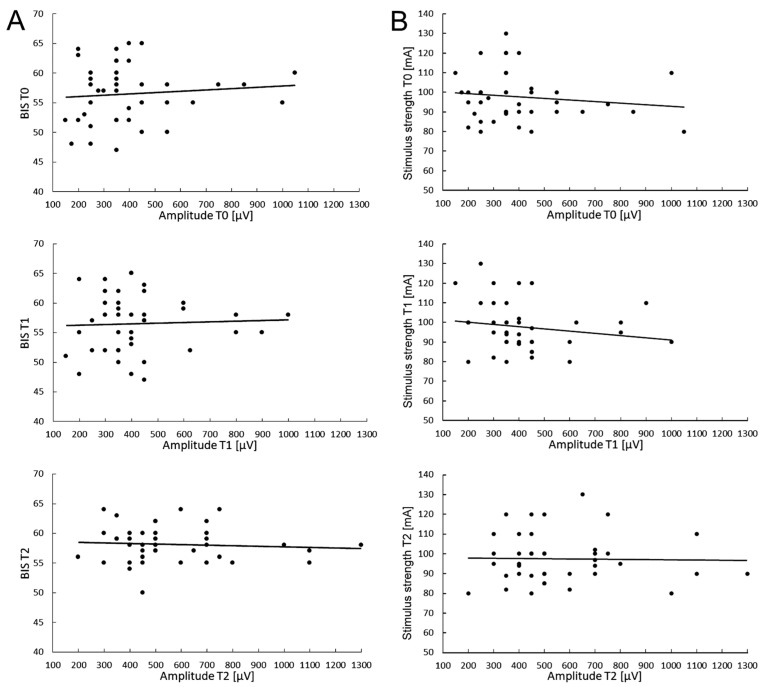
A graphical presentation of relationships between the fluctuation in the MEPs’ amplitude parameter and the applied level of anaesthesia (**A**) or the electrical stimulus strength (**B**) during surgeries of patients with IS at three observation periods.

**Table 1 jcm-12-06312-t001:** Demographic, anthropometric, and scoliosis characteristics of the patients and healthy volunteers from the control group. Minimum, maximum, mean values, and standard deviations are presented. Median values are provided in round brackets.

Variable Group of Subjects	Age (Years)	Height (cm)	Weight (kg)	BMI	Scoliosis Type [36,37]	Cobb’s Angle (Degrees) [36,37]
Patients N = 353 ♀	8–18 13.5 ± 1.8 (13)	132–183 167.3 ± 2.6 (167)	29–87 53.3 ± 5.2	17.4–29.7 22.0 ± 3.6	Lenke 1 = 90 Lenke 2 = 179 Lenke 3 = 84	Primary 41–87 57.2 ± 6.1 Secondary 31–51 38.6 ± 3.2
Healthy volunteers Control N = 80 ♀	8–17 13.9 ± 1.9 (14)	133–182 166.9 ± 2.3 (167)	28–85 53.1 ± 6.0	17.5–29.5 22.4 ± 3.5	NA	NA
*p*-value	0.243 NS	0.322 NS	0.118 NS	0.241 NS		

Abbreviations: ♀—female; NS—non-significant; NA—non-applicable; *p* < 0.05 determines significant statistical differences.

**Table 2 jcm-12-06312-t002:** Lists of events evoking the fluctuation of intraoperatively recorded MEP parameters during 353 surgeries.

Most Frequent Events during Which Warnings Were Reported to the Surgeon	Frequency of MEPs Amplitude Change	Frequency of MEPs Latency Change
Anesthesia fluctuation	45/353	15/353
Heating of spine associated with cauterization	9/353	65/353
Shocks during release of vertebral joints	55/353	5/353
Pedicle screw implantation	75/353	2/353
Corrective rods implantation	88/353	7/353
Derotation with convex rod	77/353	13/353
Distraction on concave side	66/353	16/353
Compression on the convex side	34/353	12/353

Abbreviation: MEP—motor evoked potential recorded intraoperatively.

**Table 3 jcm-12-06312-t003:** Comparison of results from electromyographical, electroneurographical, and motor evoked potential recordings performed in 353 patients pre-, intra- and postoperatively and 80 healthy volunteers (Control).

Test Parameter	Side	Control	Scoliosis Side	Patients Preoperative (1 Day before Surgery)	Control vs. Patients Preoperative	Patients Intraoperative (T0)	Patients Preoperative vs. Intraoperative (T0)	Patients Intraoperative (T1)	T0 vs. T1	Patients Intraoperative (T2)	T0 vs. T2	Patients Postoperative (1 Week after Surgery)	Control vs. Patients Postoperative	Patients Preoperative vs. Postoperative
		Min.–Max. Mean ± SD		Min.–Max. Mean ± SD	*p*-Value	Min.–Max. Mean ± SD	*p*-Value	Min.–Max. Mean ± SD	*p*-Value	Min.–Max. Mean ± SD	*p*-Value	Min.–Max. Mean ± SD	*p*-Value	*p*-Value
**Tibialis anterior muscle sEMG during maximal contraction**														
Amplitude (µV)	R	600–2600 890.6 ± 104.2	Convex	300–2200 556.3 ± 95.4	**0.041**	NA	NA	NA	NA	NA	NA	400–2200 623.5 ± 101.5	**0.041**	**0.050**
	L	600–2550 887.8 ± 91.5	Concave	200–2000 434.8 ± 88.7	**0.036**	NA	NA	NA	NA	NA	NA	200–1950 554.2 ± 99.2	**0.038**	**0.047**
*p*-value	R vs. L	0.327	Convex vs. Concave	**0.048**	NA	NA	NA	NA	NA	NA	NA	0.053	NA	NA
FI (3-0)	R	3.0–3.0 3.0	Convex	3.0–2.0 2.5 ± 0.3	**0.033**	NA	NA	NA	NA	NA	NA	3.0–2.0 2.6 ± 0.4	**0.030**	0.063
	L	3.0–3.0 3.0	Concave	3.0–1.0 2.3 ± 0.4	**0.012**	NA	NA	NA	NA	NA	NA	3.0–1.0 2.4 ± 0.4	**0.021**	0.066
*p*–value	R vs. L	NS	Convex vs. Concave	**0.044**	NA	NA	NA	NA	NA	NA	NA	**0.043**	NA	NA
**Peroneal nerve ENG recorded from extensor digitorum brevis muscle after stimulation at ankle**														
M-wave Amplitude (µV)	R	3000–12,500 6760.1 ± 965.1	Convex	1500–10,000 2725.8 ± 472.4	**0.008**	NA	NA	NA	NA	NA	NA	1400–10,500 2790.4 ± 338.5	**0.008**	0.205
	L	3000–11,600 6558.4 ± 877.3	Concave	1400–9800 2648.9 ± 584.3	**0.009**	NA	NA	NA	NA	NA	NA	1400–9950 2992.3 ± 421.9	**0.006**	**0.045**
*p*–value	R vs. L	0.228	Convex vs. Concave	0.064	NA	NA	NA	NA	NA	NA	NA	**0.053**	NA	NA
M-wave Latency (ms)	R	3.2–5.4 4.5 ± 1.1	Convex	3.3–6.2 5.1 ± 1.3	**0.041**	NA	NA	NA	NA	NA	NA	3.4–6.4 5.0 ± 1.4	**0.037**	0.171
	L	3.3–5.5 4.6 ± 1.1	Concave	3.4–6.5 5.5 ± 1.2	**0.032**	NA	NA	NA	NA	NA	NA	3.5–6.7 5.0 ± 1.2	**0.038**	**0.046**
*p*–value	R vs. L	0.328	Convex vs. Concave	**0.040**	NA	NA	NA	NA	NA	NA	NA	NS	NA	NA
F-wave Frequency (x/20 M-waves)	R	14–20 17.5 ± 1.3	Convex	10–17 12.4 ± 1.6	**0.039**	NA	NA	NA	NA	NA	NA	10–18 13.9 ± 1.5	**0.040**	0.067
	L	14–20 17.8 ± 1.4	Concave	8–16 11.2 ± 1.4	**0.034**	NA	NA	NA	NA	NA	NA	9–17 13.4 ± 1.4	**0.039**	**0.048**
*p*–value	R vs. L	0.318	Convex vs. Concave	**0.047**	NA	NA	NA	NA	NA	NA	NA	0.082	NA	NA
M-F waves Interlatency (ms)	R	38.6–49.2 44.4 ± 2.2	Convex	38.9–58.4 49.7 ± 2.5	**0.043**	NA	NA	NA	NA	NA	NA	38.6–56.1 49.1 ± 2.7	**0.045**	0.062
	L	39.0–49.4 44.7 ± 2.3	Concave	39.9–59.2 53.4 ± 3.9	**0.032**	NA	NA	NA	NA	NA	NA	38.6–57.4 49.2 ± 3.5	**0.044**	**0.047**
*p*–value	R vs. L	0.485	Convex vs. Concave	**0.041**	NA	NA	NA	NA	NA	NA	NA	0.058	NA	NA
**TMS/TES induced MEP recorded from tibialis anterior muscle**														
Amplitude (µV)	R	1200–3550 1697.2 ± 96.6	Convex	250–1300 409.9 ± 89.3	**0.009**	200 –1200 410.1 ± 94.6	0.095	300–1300 448.6 ± 72.1	0.063	500–1800 702.1 ± 82.8	**0.032**	650–2200 950.7 ± 102.5	**0.022**	**0.019**
	L	1000–2950 1609.1 ± 78.6	Concave	150–1100 379.9 ± 69.4	**0.009**	100–1000 382.4 ± 78.1	0.113	200–1000 392.5 ± 91.4	0.081	400–1350 495.9 ± 90.1	**0.045**	500–1750 806.1 ± 114.6	**0.014**	**0.016**
*p*–value	R vs. L	0.291	Convex vs. Concave	**0.049**	NA	**0.049**	NA	**0.045**	NA	**0.038**	NA	**0.041**	NA	NA
Latency (ms)	R	24.9–31.9 28.7 ± 1.3	Convex	27.9–35.8 31.8 ± 2.0	**0.032**	28.7–37.8 31.1 ± 1.8	0.157	28.9–38.1 31.9 ± 1.9	0.235	28.2–38.4 31.0 ± 1.9	0.310	28.8–39.4 31. 2 ± 2.2	**0.025**	0.064
	L	25.3–32.3 29.1 ± 1.4	Concave	28.0–37.4 32.2 ± 2.1	**0.038**	28.8–38.9 32.9 ± 2.1	0.091	29.1–39.6 33.1 ± 2.3	0.195	30.7–40.4 33.3 ± 2.2	0.372	30.9–40.1 33.4 ± 2.5	**0.036**	0.055
*p*–value	R vs. L	0.271	Convex vs. Concave	0.071	NA	0.069	NA	0.055	NA	**0.050**	NA	0.054	NA	NA

Abbreviations: T0—intraoperative observation period before surgery onset, T1—intraoperative observation period after pedicle screws’ implantation, T2—intraoperative observation period after corrective rods’ implantation, correction, distraction, and derotation of the spine curvature; sEMG—surface electromyography recordings; FI—frequency index (3-0)—frequency of motor units’ action potentials’ recruitment during maximal contraction (3—95–70 Hz—normal; 2—65–40 Hz—moderate abnormality; 1—35–10 Hz—severe abnormality; 0—no contraction); ENG—electroneurography recordings (M and F potentials); TMS—transcranial magnetic stimulation; TES—transcranial electrical stimulation; MEP—muscle-recorded motor evoked potential; NA—non-applicable; NS—non-significant; *p* ≤ 0.05 determines significant statistical differences marked in bold.

## Data Availability

All the data generated or analyzed during this study are included in this published article.

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
