# Peer review of "Comparison of Motor Evoked Potentials Neuromonitoring Following Pre- and Postoperative Transcranial Magnetic Stimulation and Intraoperative Electrical Stimulation in Patients Undergoing Surgical Correction of Idiopathic Scoliosis"

_jcm, 2023, doi:10.3390/jcm12196312_

Round 1

Reviewer 1 Report

The authors aim to compare pre- and postoperative MEP with intraoperative MEP in young patients with adolescent idiopathic scoliosis (AIS) undergoing corrective spine surgery.

It is a lengthy, but well written manuscript with an interesting topic: Comparing maximal EMG contraction, NCV studies of peroneal nerve and MEP obtained from tibial muscle pre-, intra- and postoperatively in AIS patients and investigating whether there are neurogenic abnormalities. I favor the wording “abnormalities” over injuries, as injury is suggestive for a one time impact. The authors demonstrate abnormalities in nerve conduction, F-wave and MEP studies on the concave compared to the convex side of the scoliosis interestingly those from the peroneal nerve. To follow surgery, the authors demonstrate a minimal latency reduction and increase in amplitude. This is interpreted as neuronal recovery.

The authors focus on their assumption that there is an underlying neurological disease in AIS. I do have concerns with some of those assumptions. First, there is only one reference provided supporting this idea, and this reference is an abstract only (Huber, 2012). Indeed, there are reports about EMG dysbalances in paravertebral muscles. There is discussion ever since, if there is an underlying neuromuscular disorder even in AIS. The idea of “changes in activity of grey matter” with resulting nerve conduction abnormalities is intriguing. But none, have been reported on a broad base and are not established as a neurogenic cause for AIS. The patients do not show any clinical neurological deficit. Furthermore, younger research focusses on melantonin or other growth factors being involved in the development of an AIS.

I am not aware, that there is impaired neural transmission and neurogenic abnormalities related to the scoliotic deformity. If this is the case and develops over several years, I would expect a rather slow recovery than a fast within the time-frame of a surgery. The provided amplitude differences between healthy controls and within the AIS group are difficult to understand.

Methodology:

The authors examine the peroneal nerve as an index spinal root of L5. L5 is not always involved in the curvature and is not necessarily part of screw placement and the surgical correction. In a pure thoracic curve (as shown in figure 2), I would not expect any involvement of L5 at all. Only in very severe lumbar Cobb angle resulting into neuroforaminal narrowing, I would expect a compression of the L5 spinal root. I suggest to analyze the data according to L5 involvement, this could lead to more definitive results.

Surgical procedure: The description of the surgical procedure is not needed in detail for the understanding of this study. Please shorten. An overview about the spinal levels operated on would help to understand the data. The surgical results without any neurological incidence are excellent.

TES: The use of Cz/C3 and Cz/C4 for leg muscle MEP is rather unusual. This was surely not advocated by Deletis et al. Why did you establish this stimulation electrode montage? Did you also try C3/C4, have there been differences in the stimulation intensity?

Comparing amplitude differences between TMS and TES is difficult as the train-stimulation methodology for TES results in a more disperse amplitude, which might result into amplitude differences. Although the authors consider this when discussing the reference of Lo et al., the comparison of the amplitudes is included.

Results

Table 1: please provide the median of age and of height. A higher median in scoliotic patients might explain the differences.

Figure 2: The shorter latency between the pre and post-condition is not really convincing. The stimulation artefact of the TES MEP alters? Was the stimulation polarity or the intensity changed?

There are manifold influences on TES MEP amplitudes. An increment of amplitudes in a scoliosis surgery of long hours is usually an effect of anesthesia. In my experience, I rarely see increments of amplitudes and if so, those are usually related to a lighter anesthesia or an elevated blood-pressure. I would be very, very cautious to claim it an effect of surgical deformity correction. Despite a steady state in BIS, there is mainly an accumulation of Propofol ® and such “deeper” anesthesia over time. The authors did not experience MEP fading, but the vice versa. This indeed is rather unusual and should be discussed.

Table 3: please provide the names of the tests within the table. Maybe you did, but it does not show in my version.

From the point of neurophysiology, the reported latency shifts of 0.1 – 0.5 ms are within the range of the normal inter-individual variability of the method. A sound change would be a difference of double standard-deviation and a normalization within this range. It would be more precise, to compare pre- and postoperative values with values obtained from healthy controls at the same time interval.

It is difficult to understand the longer latencies of the F-waves in the patients’ group. Have there been outliers explaining these findings? Would this correlate to L5 involvement? As given, there is not any obvious difference in the latencies between the concave and convex side, and the statistical effect might be related to the large amount of data.

Discussion and conclusion:

Indeed, the amplitude increments in the scoliosis patients are difficult to understand, but the explanation given on page 16 reads very speculative. Relating amplitudes obtained from MEP studies in tibial anterior muscle, which was intended to assess spinal root L5, with “spinal roots compressed in narrowed intravertebral foramina” cannot be performed in general way. First, the curvature must be high to result into a compressed neuroforamina, second why should a “compressed thoracic neuroforamen” result into an alteration of the L5 root?

The statement that it is the first time, that pre-, intra- and postoperative MEP shows to be effective to prevent neuro-deficits, cannot be made and is not supported by the methodology and study results. Indeed, it is the first study to compare all three time points in scoliotic patients. But, the effectiveness with regard to prevention of neuro-deficits was neither the intention, nor the focus of the study and such was not shown. The surgical results are excellent, as neither MEP loss, nor neurological deficits occurred, but this might also happened without any monitoring after all. From the study, I cannot see any use of the pre- and postoperative recordings for prevention of neuro-deficits.

Overall, the manuscript is lengthy and elaborates on many ideas. I suggest to focus on the comparisons between pre-, intra- and postoperative results and analyze those with regard to curvature and involvement of the spinal root L5.

Reviewer 2 Report

I read the manuscript „Comparison of motor evoked potentials neuromonitoring fol-2 lowing pre- and postoperative transcranial magnetic stimulation and intraoperative electrical stimulation in patients undergoing surgical correction of idiopathic scoliosis“ by PrzemysÅ‚aw Daroszewski and colleagues.

This lengthy and difficult-to-read manuscript intends to give evidence that surgical correction of scoliosis improved motor evoked potentials monitored intraoperatively and elicited by  TES vs. pre and postoperative testing by TMS.

 I have a couple of objections authors have to answer to proceed with possible publications of their work.

Authors hypothesize that correction of scoliotic curvature  „improve spinal neural transmission“  (lines 30,34, 35 of their abstract). This conclusion is pulled from data on the parameters of MEPs improved at the convex side of curvature measured by the amplitude of motor-evoked potentials in the tibial anterior muscle.

I am not convinced that their offer sufficient data to give evidence that improving parameters of MEPs after correction of scoliosis is only due to „improve spinal neural transmission“, a rather vague description for the improved conductivity through descending tract and neural structures generating MEPs.

Many other mechanisms could be explanations for their data. One of them is the asymmetry of the tendon reflexes recorded in the paraspinal muscle reflecting alfa motor neuron excitability, possibly involving other segments of the scoliotic spinal cord. Trontelj JV and all., have elaborated on this a long time ago, giving evidence of segmental neurophysiological mechanisms in scoliotic patients.

This and other possible neurophysiologic mechanisms should be discussed to explain their data. The mechanical correction of a scoliotic spine can influence this segmental mechanism.

Keywords: „Postoperative neuromonitoring „ is incorrect. The authors postoperatively did not monitor but made control measurements.

Lines 162 -164 and throughout the text are incorrect, claiming that MEPs measure „spinal cord function“ or neurological status. MEPs measure the functional integrity of the structures that generate the motor-evoked potentials.

Fig. 1 “rubrospinal tract” There is no evidence that  at least in the men, the rubrospinal tract contributes to the generating MEPs

Lines 357-358 Stimulating electrodes were positioned over the skull using the 10 system: Cz-C3 3-6 cm to the left and Cz to C4 – distance 3-6 cm to the right [Deletis 2007… First, that is Deletis et all., the wrong citation, it was not such a recommendation in the cited paper. The second C3 or C4 does not correspond to the 3-6 cm lateral to the Cz. Usually is 6-7 cm lateral to Cz, never 3 cm lateral to Cz.

Lines 389-391, 437-438 “Pilot observations indicated that overheating of the tissues accompanying the cauterization before T1 could affect the conduction of nerve impulses in the spinal cord pathways within the white matter funiculi.” There is no evidence for that. The spinal cord is rather thermically isolated by its bony structure preventing overheating by monopolar cautery. I believe that author observe a coincidence of changes in the parameters of MEPs but could be another, highly probably, technical not biological, reason for that.

In their figure 2, (above) I superimposed preoperative MEPs in red over postoperative MEPs in grey (trace E is superimposed over trace G in their figure). The latencies of the beginning of both responses are identical, while their peak latencies are significantly longer for postoperative recordings. How did the authors explain that?
